biophysics/spectroscopy

human bones, moderate heating, aerobic versus anaerobic heating, FTIR-ATR, INS, forensic and archaeological sciences

**Authors for correspondence:**
L. A. E. Batista de Carvalho
e-mail: labc@ci.uc.pt
S. F. Parker
e-mail: stewart.parker@stfc.ac.uk

This article has been edited by the Royal Society of Chemistry, including the commissioning, peer review process and editorial aspects up to the point of acceptance.

# The impact of moderate heating on human bones: an infrared and neutron spectroscopy study

M. P. M. Marques[1,2], L. A. E. Batista de Carvalho[1], D. Gonçalves[3,4,5], E. Cunha[2,3] and S. F. Parker[6]

[1]Molecular Physical-Chemistry R&D Unit, Department of Chemistry, University of Coimbra, 3004-535 Coimbra, Portugal
[2]Department of Life Sciences, [3]Laboratory of Forensic Anthropology, Centre for Functional Ecology, and [4]Research Centre for Anthropology and Health (CIAS), University of Coimbra, 3000-456 Coimbra, Portugal
[5]Archaeosciences Laboratory, Directorate General Cultural Heritage (LARC/CIBIO/InBIO), 1349-021 Lisbon, Portugal
[6]ISIS Facility, STFC Rutherford Appleton Laboratory, Chilton, Didcot OX11 0QX, UK

MPMM, 0000-0002-8391-0055; LAEB, 0000-0002-8059-8537; SFP, 0000-0002-3228-2570

This study aims to analyse human bones exposed to low/medium temperatures (200–650°C) under experimentally controlled conditions, both oxidizing and reducing, using complementary optical and neutron vibrational spectroscopy techniques. Clear differences were observed between the aerobically and anaerobically heated bones. The organic constituents disappeared at lower temperatures for the former (*ca* 300°C), while they lingered for higher temperatures in anaerobic environments (*ca* 450–550°C). Unsaturated non-graphitizing carbon species (chars) were detected mainly for anaerobically heated samples, and cyanamide formation occurred only at 650°C in reducing settings. Overall, the main changes were observed from 300 to 400°C in anaerobic conditions and from 450 to 500°C in aerobic environments. The present results enabled the identification of specific spectroscopic biomarkers of the effect of moderate temperatures (less than or equal to 650°C) on human bone, thus contributing to a better characterization of forensic and archaeological skeletal remains subject to heating under distinct environmental settings. In particular, these data may provide information regarding cannibalism or ancient bone boiling and defleshing rituals.

# 1. Introduction

Bone is a highly heterogeneous material, comprising an organic part (*ca* 20 wt%, mainly type I collagen and lipids) interweaved in an inorganic matrix of carbonate-substituted hydroxyapatite ($Ca_{10}(PO_4)_6OH_x$, HAp, *ca* 70 wt%), and water (*ca* 10 wt%) [1]. When subject to heat, bone undergoes changes according to four consecutive main stages, which are accompanied by macroscopic variations in colour, dimensions, weight, porosity and crystallinity. The stages are: (i) dehydration—removal of water by evaporation—at a temperature range of 100–500°C; (ii) decomposition—loss of the organic components (lipids and collagen), which is usually associated with a significant colour change—at a temperature interval of 200–600°C (depending on oxygen levels and amount of soft tissue); (iii) inversion—changes and reorganization within the inorganic matrix (mainly regarding the carbonates and phosphates)—at 500–1100°C; (iv) fusion (microcrystallinity variations)—crystals within the bone's framework melt and coalesce, with concomitant changes in dimensions—at a temperature range of 700–1200°C. These are not discrete phases—a given bone may experience more than one process simultaneously, in different parts of the bone. Although there is an expected weight loss immediately after starting the heating process (becoming stable at 500°C at *ca* 60% of the original value) [2,3], no measurable reduction in volume has been observed up to 650–700°C (after which it decreased steadily). Apart from the temperature and the duration of heat exposure, other factors can determine the impact of heat on bone tissue, namely the available oxygen (aerobic or anaerobic settings) and the characteristics of the surrounding medium (e.g. the presence of metals or contaminant organic materials), as well as the morphology of the bone (e.g. ratio of compacta to spongiosa structures) [4,5].

The identification of cooked human bones found in archaeological sites may deliver information regarding funerary rituals (such as bone boiling and defleshing after death) or help to establish cannibalism. Additionally, accurate structural and chemical data retrieved from modern burned bones may be essential for victim identification—as bones subject to heating above 250–300°C often do not retain DNA capable of being amplified, or this entails a high risk of contamination that hinders DNA typing [2,6,7]. Burned skeletal remains found in archaeological or forensic settings may not always have been subject to intense heating (greater than 600–650°C). Instead, the bone may have undergone an indirect thermal exposure, which is often the case in archaeological sites or in forensic scenarios when criminal burning of cadavers takes place, due to the insulating effect of the flesh surrounding the bone (the temperature that it experiences being significantly reduced). Bone subject to heating below *ca* 500°C has a higher porosity and is therefore more fragile and not so well preserved in the soil as high temperature burned bone.

Reports on the analysis of low and medium temperature heated bones (less than or equal to 650°C) are scarce, only a few studies having been published using physicochemical techniques such as surface and transmission electron microscopy and X-ray methods [8–10], or less reliable colorimetric analysis [11,12]. Very few spectroscopic studies are to be found on these types of samples (only faunal bones) [8,13], as opposed to heavily burned bones (650 to 1000–1100°C) which have been investigated by the authors and several others through vibrational spectroscopy in the last few years [14–28]. Research on moderately heated bone is complicated by the fact that diagenetic processes (physical and chemical changes occurring after death) can mimic the effects of low intensity heating, mainly upon long-term burial. In some cases it has been impossible to distinguish between buried and moderately heated bone [13]. Additionally, low to medium heating does not cause significant changes in bone's mineral components, which leaves the variations in the organic constituents (collagen and lipids) as the only biomarkers of the impact of temperature, with an emphasis on collagen fibril unpacking and fragmentation which arise well before collagen loss and change in mineral crystallinity (that take place at higher temperatures). Hence, spectroscopic relationships such as the crystallinity index (splitting factor), or the carbonate/phosphate and OH/phosphate ratios, which were reported to be highly valuable in the analysis of heavily burned bones [15,17,20–22,27], are useless in the characterization of bones subject to temperatures below 600°C.

Fourier transform infrared (FTIR) spectroscopy has been shown to be a technique of choice for the study of bone tissue, even upon extensive burning, since it allows a molecular level analysis with high accuracy and sensitivity, providing reliable information on the heat-induced chemical and structural alterations [15,18–20,24,28–31]. In particular, FTIR in attenuated total reflectance (ATR) mode, avoiding any type of sample preparation and requiring minimal amounts of bone, has become the most commonly used, rapid and non-invasive, spectroscopic tool for the analysis of skeletal remains, both recent [23,24,28,32] and archaeological [17,21,26,33–35]. Inelastic neutron scattering (INS) is a non-optical vibrational spectroscopy technique very suitable for probing bone since it is highly sensitive to

hydrogen. In addition to having no selection rules, it allows access to the very low energy range of the spectrum (0–400 cm$^{-1}$) with high sensitivity, thus enabling the detection of vibrational features arising from bone's inorganic framework [18,24,28]. A few initial reports on the application of INS to the analysis of unburned faunal bone [36–38] were later extended by the authors to human burned skeletal remains [18,21,22,24,26,28,35,39]. (A more extensive description of INS is given in the electronic supplementary material.)

In the present work, we build upon our successful studies of human skeletal remains subject to intense heating (greater than 600°C). Here, we have generated a new set of samples that focuses on the (relatively) unexplored low temperature range: 200 to 650°C. The same type of human bones (femur and tibia) are used, under experimentally controlled conditions, both aerobic (combustion, in an oxidative medium) and anaerobic (reductive medium, in a closed chamber), probed by complementary optical and neutron vibrational spectroscopy—FTIR-ATR and INS. The lowest temperature studied has been reduced to 200°C to include temperatures that are easily achieved in cooking vessels. The results provide valuable data on moderately burned human bones (regarding temperature and environmental parameters such as oxygen availability), allowing the re-creation of the bone heating conditions and thus to obtain relevant archaeological information and assist forensic investigation.

# 2. Material and methods

The material and methods employed are similar to those used previously [18,21,22,24,26,28,35,39], so only a summary is provided here. More detailed information on the materials, sample preparation, FTIR-ATR acquisition, and INS fundamentals, acquisition and analysis can be found in the electronic supplementary material.

## 2.1. Materials

The bone samples were collected from a femur and a tibia of a human skeleton belonging to a collection of unidentified human skeletons donated for research purposes, hosted at the Laboratory of Forensic Anthropology of the University of Coimbra [40].

## 2.2. Sample preparation and experimental burning

Thirty-two samples were used in this study: 16 from the femur and 16 from the tibia. Each experimental burning included one femur and one tibia fragment aerobically burned, as well as one femur and one tibia sample anaerobically burned, as previously described [28] (electronic supplementary material, figure S1).

The following maximum temperatures and durations were applied (for a heating rate of $ca$ 6–10°C min$^{-1}$): 200°C (120 min), 300°C (120 min), 400°C (120 min), 450°C (120 min), 500°C (120 min), 550°C (120 min), 600°C (120 min) and 650°C (120 min). These burning durations reflect the time required to attain each maximum temperature, after which the furnace was switched off. Before removal from the furnace, the samples were left to cool to room temperature. After burning, each bone fragment was ground and sieved (to a mesh size of 400 µm), yielding 2 to 3.5 g.

## 2.3. Vibrational spectroscopy

Infrared spectra were recorded at room temperature in $CO_2$-free dry air. The spectra were normalized to the $v_3(PO_4)$ band at 1030 cm$^{-1}$.

Parallel INS measurements in TOSCA [41–44] and MAPS [42,45,46] provide all the vibrational region of interest with high sensitivity (from very low wavenumbers to the OH stretching interval, 8–4000 cm$^{-1}$). All measurements were carried out at 5–10 K, in order to reduce the impact of the Debye–Waller factor (exponential term in equation (1), electronic supplementary material).

# 3. Results and discussion

Human femur (F) and tibia (T) were analysed upon aerobic and anaerobic burning at temperatures between 200°C and 650°C. In order to avoid inter-skeleton variability, the samples were collected

from the same skeleton [20] and two types of bones were probed in order to eliminate potential replicability problems.

In real scenarios, bone is covered with a thick soft tissue layer during the early stages of burning, which has a protective effect against external insults such as heating. Hence, it is important to understand how moderate temperatures and reduced oxygen availability (anaerobic media) affect the outcome of the burning process. Different thermal alteration profiles have been reported for bone burned under either aerobic or anaerobic conditions, both for faunal [11,25,31] and human skeletal remains [28] subject to high temperatures (500–1000°C). These have revealed a retarding effect on the loss of the organic constituents in reducing environments. This is the first study, at the molecular level, on low and medium intensity heated bone (less than 500°C). In the present work, two different settings were probed: (i) combustion, in the presence of oxygen (oxidizing conditions)—aerobic (*A*) and (ii) absence of oxygen (reducing conditions)—anaerobic (*An*)—in a sealed chamber not allowing the release of the volatiles formed during the burning process, which enables a re-equilibrium to be attained within the bone matrix.

The bones heated in these distinct environments displayed a marked macroscopic variability regarding their colour. In agreement with previous reports [12,31,47], those burned aerobically showed a colour sequence with increasing temperatures from black (at 200°C) to greyish (at 300°C) and brown (at 400–650°C). The samples heated anaerobically (in a sealed container) were brown to dark brown at the lowest temperatures (200–400°C) and consistently black above 500°C due to the formation of amorphous inorganic carbon (charring), which is not produced in the presence of oxygen (when volatile $CO_2$ is formed instead). These differences already constitute macroscopic evidence of distinct thermal-induced changes within the bone, in either aerobic or anaerobic settings [11].

Upon a temperature increase, heat-degradation events such as water and carbonate loss, destruction of the organic constituents (lipids and collagen) and ultimate crystallinity rearrangements were evidenced through the vibrational profile of the samples. Regarding the impact of heat on their vibrational signature (infrared and INS), no significant variations were observed between the two types of bone under analysis (femur and tibia), for each experimental condition, as expected based on previous reports [24,28].

## 3.1. Heat impact on bone's organic components

A strikingly different impact of heat on bone's organic constituents was found between aerobic and anaerobic burning environments, supporting previous observations for bone heated at temperatures above 500°C [28]. In the presence of oxygen, the lipids were destroyed between 300 and 400°C, as evidenced by the disappearance of the $\delta(CH_2)$ bands at *ca* 1450 cm$^{-1}$ and $\nu(CH_2)$ at *ca* 3000 cm$^{-1}$ in the INS signatures (figures 1–3). The proteins (mainly collagen) were completely degraded at 400–450°C (loss of the amide I, $\nu$(C=O), signal at *ca* 1650 cm$^{-1}$, figures 3 and 4). Under reducing conditions, a retarded effect was observed: the lipids were present until 450–500°C and the collagen amide I mode was seen up to 600–650°C (figure 3). Also, the protein amide II signal at *ca* 1550 cm$^{-1}$ ($\delta$(N–H)+$\nu$(C–N)) was detected by infrared spectroscopy at 200°C under aerobic conditions but up to 400°C in anaerobic settings (figure 4). The INS methyl torsion signal from the polypeptide chains ($\tau(CH_3)$ at 250 cm$^{-1}$), in particular, is a good indicator of the heat-elicited degradation of protein components: under oxidative conditions it was clearly detected only at 200°C (figure 3a), while for anaerobically burned samples it was observed up to 450°C (figure 3b).

The INS feature at *ca* 2980 cm$^{-1}$ (ascribed to $\nu(CH_2)_{sp3}$ from lipids and proteins) gradually disappeared, giving rise to a signal at 3090–3100 cm$^{-1}$, at 300–400°C and 450–500°C respectively for aerobically and anaerobically burned samples (figure 1; electronic supplementary material, figure S2). This is directly correlated with lipid degradation, as also evidenced by the disappearance of the corresponding $\delta(CH_2)$ INS bands. Additionally, the vibrational feature at higher wavenumbers (*ca* 3100 cm$^{-1}$) detected for bones heated under reducing conditions (figure 1) is ascribed to the $\nu$(CH) mode of edge-terminating hydrogens from non-graphitizing carbon species (char, comprising disorganized sp$^2$ carbon), which are known to be formed upon heating in anaerobic environments. This band, previously detected by INS in glassy carbons [48], was observed above 500°C (up to 650° C) for all the anaerobically burned samples, but not for those burned in the presence of oxygen.

The very broad infrared bands detected between 3100 and 3500 cm$^{-1}$ are due to NH and OH stretching vibrations (figure 4). Those assigned to protein NH groups (amide A band, coexisting with amide I at 1650 cm$^{-1}$) were observed at 3280 cm$^{-1}$ for the samples heated up to 200/300°C aerobically, and up to 600°C anaerobically. Above these temperatures, for each burning environment,

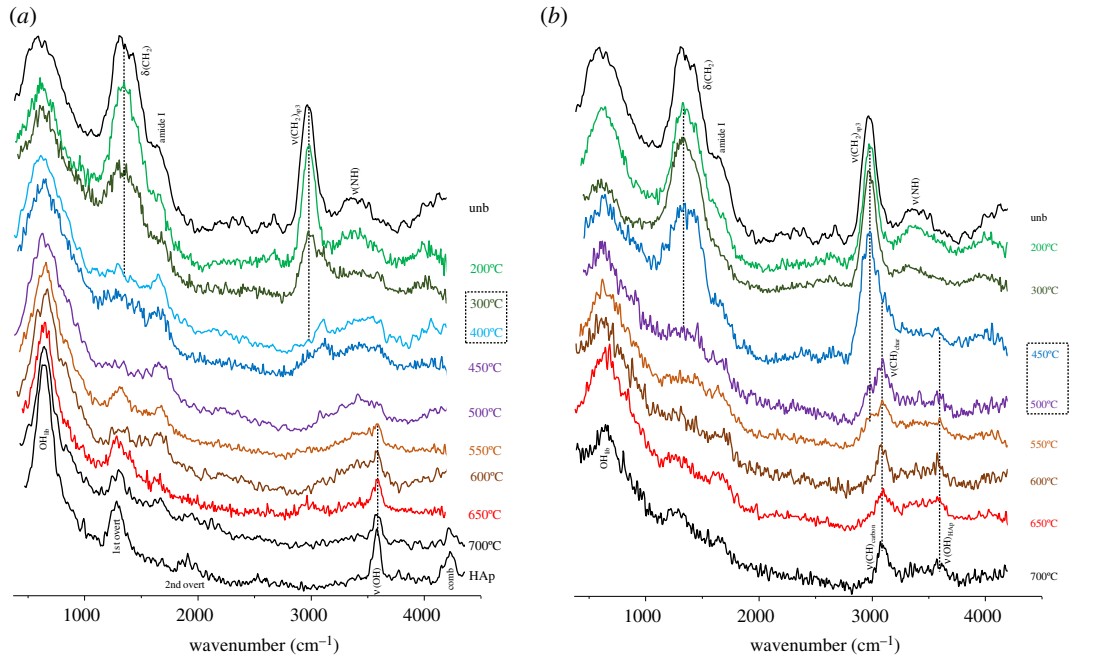

**Figure 1.** INS spectra (measured in MAPS, with 5240 cm$^{-1}$ incident energy) of human femur burned at temperatures between 200 and 650℃, either under aerobic (*a*) or anaerobic (*b*) conditions. The temperature range corresponding to the largest spectral variations is highlighted by a dashed rectangle. (The spectra of reference calcium hydroxyapatite (HAp, SRM 2910b) and of femur unburned (unb) and burned at 700℃ [28] are also shown, for comparison purposes.)

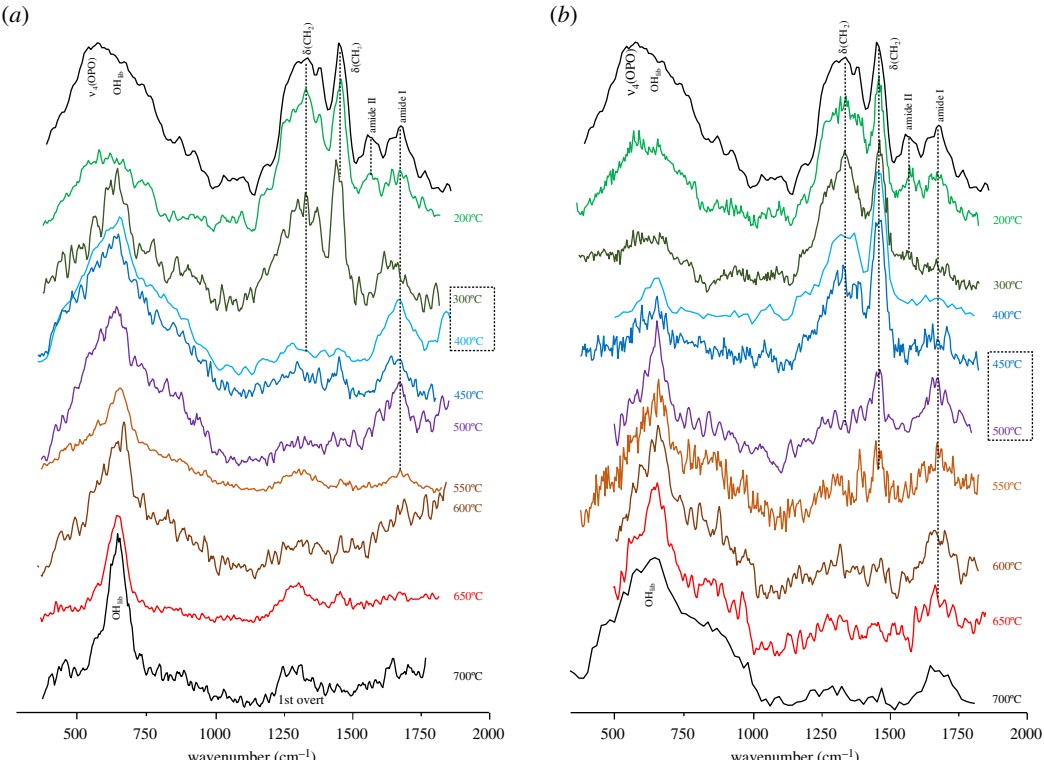

**Figure 2.** INS spectra (measured in MAPS, with 2024 cm$^{-1}$ incident energy) of human tibia burned at temperatures between 200 and 650℃, either under aerobic (*a*) or anaerobic (*b*) conditions. The temperature range corresponding to the largest spectral variations is highlighted by a dashed rectangle. (The spectra of tibia unburned (unb) and burned at 700℃ [28] are also shown, for comparison purposes.)

this feature was gradually substituted by a signal at 3360 cm$^{-1}$ ascribed to amine moieties produced by degradation of the polypeptide chains, as well as to OH groups arising from HAp's disruption upon heating.

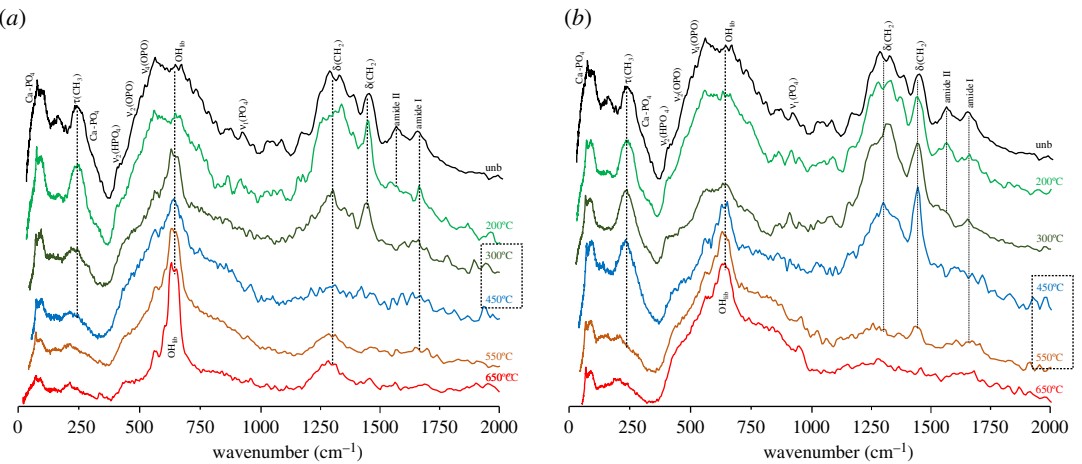

**Figure 3.** INS spectra (measured in TOSCA, 0–2000 cm$^{-1}$) of human femur burned at temperatures between 200 and 650°C, either under aerobic (*a*) or anaerobic (*b*) conditions. The temperature range corresponding to the largest spectral variations is highlighted by a dashed rectangle. (The spectrum of femur unburned (unb) [28] is also shown, for comparison purposes.)

The distinctive differences in heat-elicited effects on bone, in either aerobic or anaerobic settings, are more clearly seen when comparing the corresponding vibrational signatures at some illustrative temperatures (figure 5). The infrared profiles at 300 and 650°C visibly showed the decreased amount of protein for aerobically burned samples as compared to those burned under reducing conditions (at 300°C), alongside with the appearance of characteristic bands from the hydroxyapatite framework (e.g. OH libration) for the former (figure 5*a*). In turn, analysis of the INS spectra revealed that the CH$_2$ deformation and stretching modes from lipids and proteins (respectively at 1350–1450 and 2980 cm$^{-1}$) were still observed at 450°C under anaerobic conditions, disappearing only at *ca* 550°C, as opposed to aerobic heating for which these vibrations were already lost at 450°C (figure 5*b,c*). For the lowest temperature currently probed (200°C) both the INS and FTIR profiles were found to be similar, irrespective of the burning conditions regarding oxygen availability (figures 4 and 5*b*). These results are in agreement with macroscopic changes previously observed in combusted bones, namely the progressive loss of water and organic material starting almost immediately after aerobic heating, responsible for a marked decrease in bone weight that reached a plateau at 450–500°C (60% of the original value) [2,49]. The information presently gathered is also in accordance with former studies on moderately heated faunal bones in oxidative environments (by thermogravimetry and differential thermal analysis), which reported lipid destruction starting around 45°C, followed by dehydration at 100°C (evaporation of free water within the bone matrix) and collagen denaturation from 237°C (triple helix to random coil transition) to 327°C (loss of mechanical integrity) [50,51].

## 3.2. Heat impact on bone's inorganic matrix

While the main changes in bone's organic composition take place immediately after starting the heating process, the crystallinity rearrangements within hydroxyapatite occur mainly above 600°C. The characteristic vibrational features from bone's inorganic framework (HAp), namely the OH librational (OH$_{lib}$, at 640–660 cm$^{-1}$) and stretching ($v$(OH), at 3570 cm$^{-1}$) modes, were detected by FTIR and INS (the latter being significantly more sensitive for the libration), and were found to become more defined as the temperature increased. This is due to an enhanced crystallinity, linearly related to protein degradation [13], and is reflected in the progressive narrowing of the vibrational features particularly the infrared $v_3$(PO$_4$) and INS OH$_{lib}$ bands (figures 1*a* and 4). This effect is much more noticeable for aerobically burned samples, as clearly evidenced by the INS profiles of combusted versus reductively burned bones (figure 1), the latter containing a higher amount of amorphous hydroxyapatite and even different HAp polymorphic structures (either hexagonal or monoclinic [52]). $v$(OH)$_{HAp}$ was observed by infrared spectroscopy as a very narrow band at 3570 cm$^{-1}$, visible at 300°C and above for both aerobic and anaerobic settings, and giving rise to an intense and very defined peak at 600 and 650°C under oxidative conditions (as opposed to reducing medium) (figure 4). Above 300°C or 450°C respectively for either aerobically or anaerobically heated samples, an IR band at 3360 cm$^{-1}$ was observed comprising $v$(OH) modes from loosely bound hydroxyls

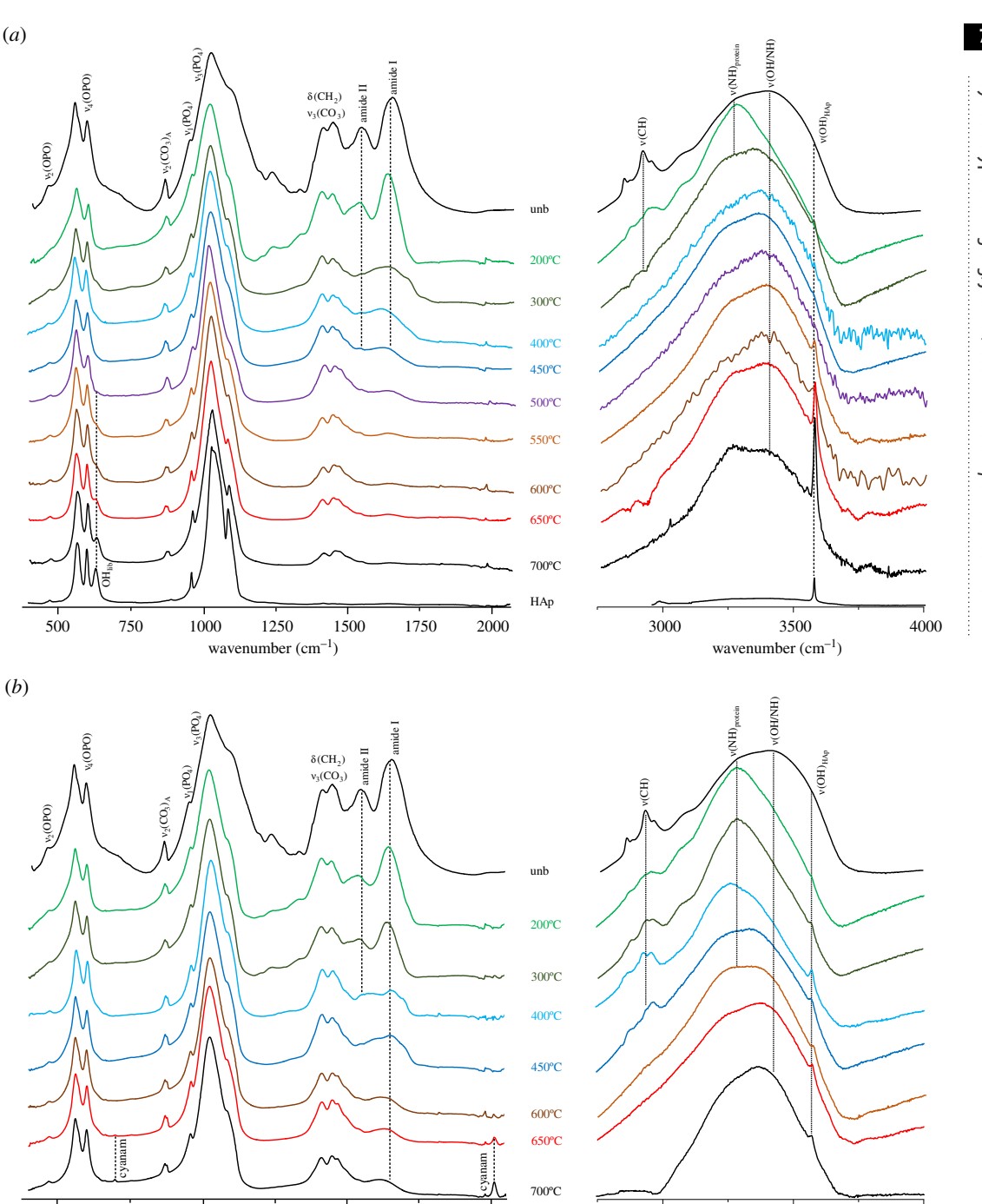

**Figure 4.** FTIR-ATR spectra (400–4000 cm$^{-1}$) of human femur burned at temperatures between 200 and 650°C, either under aerobic (*a*) or anaerobic (*b*) conditions. (The spectra of reference calcium hydroxyapatite (HAp, SRM 2910b) and of femur unburned (unb) and burned at 700°C [28] are also shown, for comparison purposes.)

resulting from heat-prompted disorganization of the hydroxyapatite framework (apart from $v$(NH) due to amines from protein degradation, as discussed above). It is known that hydroxyl groups within bone's inorganic lattice display a high mobility within the crystalline network at high temperatures, being frequently exchanged by other anions (such as Cl$^{-}$, F$^{-}$ or CO$_3{}^{2-}$, during diagenesis) [53,54]. Hydroxyapatite's OH$_{lib}$ signal, in turn, was very clearly detected by INS (highly sensitive to these types of vibrations), yielding an intense band for all temperatures and conditions which turned into a well-defined and narrower signal at 650°C for oxidative heating conditions (figures 2 and 3). The corresponding infrared feature was only seen (with low intensity) above 500°C for the aerobically burned samples and was virtually undetected for the anaerobically heated bones (figure 4), which

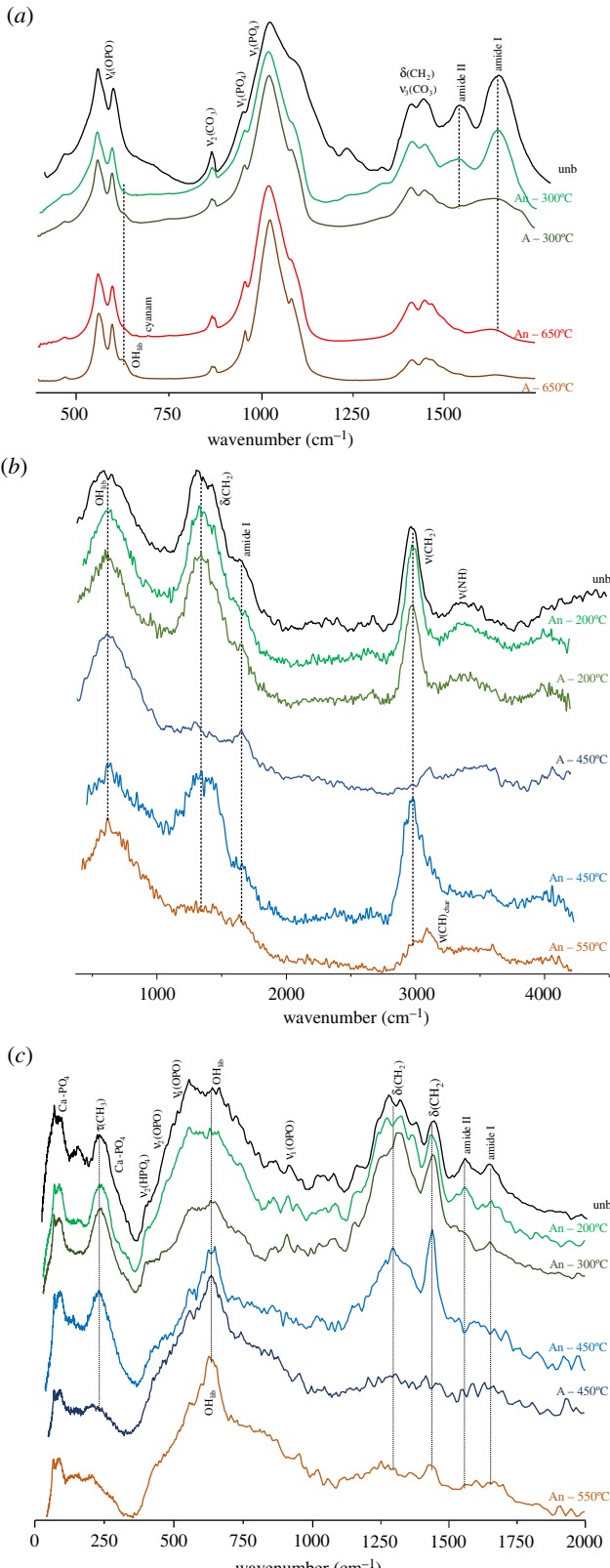

**Figure 5.** Spectra of human tibia burned at temperatures between 200 and 650°C, either under aerobic (*A*) or anaerobic (*An*) conditions: FTIR-ATR (*a*); INS, measured in MAPS (with 5240 cm$^{-1}$ incident energy) (*b*); INS, measured in TOSCA (*c*). (The spectra of tibia unburned (unb) [28] are also shown, for comparison purposes.)

may be explained by the small dipole moment variation associated with this lattice mode. Additionally, the resonance between OH$_{lib}$ and $\nu_4$(OPO) which was formerly detected for combusted bones [28] does not occur for the reductively heated samples, probably because for the different HAp polymorph present

in the latter the librational band is blue-shifted thus hindering the resonance effect (and contributing to its very low infrared intensity). The detection of the hydroxyl's vibrational signals in the currently analysed bones burned anaerobically within a closed chamber (not allowing venting of volatile products) is a clear evidence that a re-hydroxylation process takes place in this temperature range (200–650°C), analogously to what was previously observed for higher temperatures (600–1000°C) [28]. This re-equilibrium enables the hydroxyl groups that are initially released from bone's inorganic lattice upon heating to be incorporated back into this framework.

Regarding the carbonate bone components, the typical $\nu_3(CO_3)$ mode at *ca* 1450 cm$^{-1}$ was observed by infrared spectroscopy up to 300°C under oxidative conditions and up to *ca* 600°C in reductive media (figure 4). Carbonates (mostly type B) were therefore found to linger within the bone up to higher temperatures for anaerobically burned samples, which was also evidenced by the infrared $\nu_2(CO_3)$ feature (at 870–875 cm$^{-1}$) that gave rise to slightly higher intensity bands for the bones heated in reducing settings (figure 4). Since type A carbonates ($CO_3^{2-}$ for $OH^-$ substitution within bone's matrix) are less common than type B ($CO_3^{2-}$ for $PO_4^{3-}$ substitution), and the $CO_3^{2-}$ anion is considerably larger than $OH^-$, the fomer are more labile and so they are expected to be removed at an earlier stage of the heating process.

Former studies of human skeletal remains burned at high temperatures under anaerobic conditions revealed the presence of cyanamide (N≡C–NH$_2$) at 700°C or above [17,25,28], alongside a heat-triggered phase transformation of bone's hydroxyapatite framework (yielding new HAp polymorphic species). Cyanamide formation (in the presence of free ammonia formed during the burning process) has been ascribed to an incomplete oxidation of organic matter, and leads to incorporation of $CN_2^{2-}$ ions into bone's inorganic matrix substituting for $OH^-$, yielding cyanamidapatite (Ca$_{10}$(PO$_4$)$_6$(CN$_2$)$_4$) [31,47,54,55]. This process was formerly reported for ammonia-treated hydroxyapatite heated at 900 to 1200°C [54]. For the burning conditions currently used no cyanamide was detected, except for the sample subject to the highest tested temperature (650°C) in a reducing medium. In this case, the sharp infrared features from $CN_2^{2-}$ at 700 cm$^{-1}$ (N–C≡N bending) and 2009 cm$^{-1}$ (antisymmetric C≡N stretching) [58,60] were observed with very low intensities (figure 4). This allowed us to identify 650°C as the lower limiting temperature for cyanamide production upon anaerobic bone burning and subsequent nitridation of the bone matrix, no such process taking place below 650°C, nor for the aerobically heated samples. Furthermore, due to $OH^-$ by $CN_2^{2-}$ substitution following cyanamide formation, the bands characteristic of HAp's hydroxyls (namely $\nu(OH)$ at 3570 cm$^{-1}$) displayed a much lower intensity for heated samples in a reducing atmosphere, as evidenced when comparing the IR spectra at 650°C for anaerobic and aerobic burned bones (figure 4*a*,*b*, respectively). Once the species present in bone upon cyanamide formation is the $CN_2^{2-}$ anion (not containing hydrogen atoms), no signals from cyanamide were detected in the INS spectra of the sample burned at 650°C under reducing conditions. In fact, the intense INS features characteristic of N≡C–NH$_2$ arise from the vibrational modes associated with the NH$_2$ moiety (which is lacking in the anionic form)—lattice modes, $\delta(NCNH_2)$, $\delta(NH_2)_{wagging}$ and $\delta(NH_2)_{twist}$ at 178, 440, 566 and 1172/1564 cm$^{-1}$, respectively (electronic supplementary material, figure S3), in agreement with previously reported Raman and infrared data [56].

## 3.3. Quantitative effects of the heating process on bone's vibrational profile

Table 1 summarizes the main variations detected by INS and FTIR for the human bones burned either aerobically or anaerobically at temperatures below 650°C.

The temperature dependence of the hydroxyl libration wavenumber measured for the bones heated between 200 and 650°C (aerobically and anaerobically) is presented in figure 6*a*,*b*, which also encompasses the results from previous studies for higher temperature burning [24,28]. Interestingly, for oxidizing heating conditions the trend observed for the lower temperature interval (200 to 600°C) was inverse to the one detected for the higher temperature range (650–1000°C). For these higher temperatures (greater than or equal to 650°C), the profile for aerobically burned samples revealed a shift to lower frequencies with increasing temperature (figure 6*a*), while that for anaerobically heated bones showed a deviation to higher wavenumbers, following the tendency already revealed below 650°C (figure 6*b*) [28]. In addition, the variation measured for HAp's hydroxyl libration was found to be more significant below 650°C than above this temperature, mainly for aerobically heated samples: respectively $\Delta = 27$ cm$^{-1}$ versus 9 cm$^{-1}$ for aerobic media, and $\Delta = 30$ cm$^{-1}$ versus 23 cm$^{-1}$ for anaerobic settings. The variation of the collagen amide I infrared intensity with burning temperature (up to 650°C) is shown in figure 6*c*, evidencing the faster protein degradation under aerobic heating

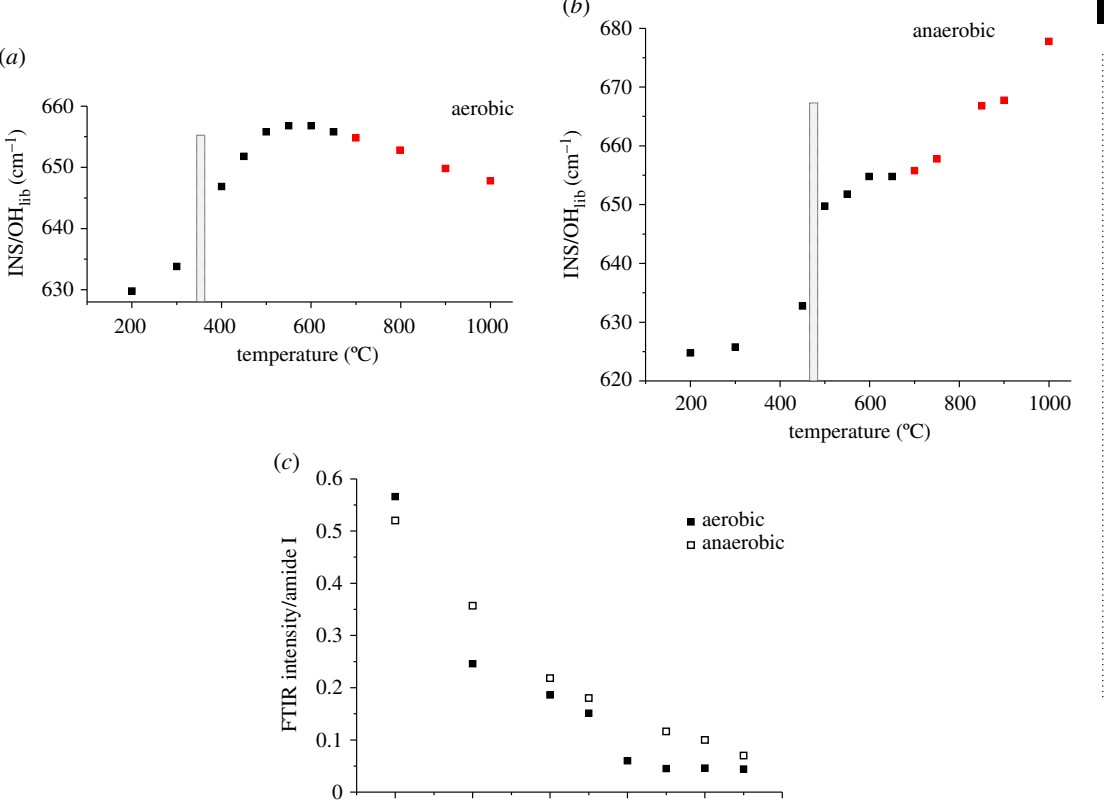

**Figure 6.** Plots of temperature dependence of the INS frequency of the hydroxyapatite's hydroxyl libration (*a,b*), and the infrared intensity of the amide I band (*c*), for human femur burned under aerobic or anaerobic conditions. (INS data measured in MAPS, with 2024 cm$^{-1}$ incident energy. The values represented with red dots are from [24] and [28]. In (*a,b*), the vertical bars highlight the largest variation of OH$_{libration}$ frequency with temperature.)

**Table 1.** Main changes observed by INS and FTIR for human bones burned under different settings—aerobic or anaerobic (sealed chamber)—at temperatures between 200 and 650°C.

| vibrational band | wavenumber (cm$^{-1}$) | observed by | burning conditions | |
|---|---|---|---|---|
| | | | aerobic | anaerobic |
| $\tau(CH_3)_{protein}$ | 250 | INS | 200°C | $\leq$450°C |
| $(OH)_{lib}$ | 640–660 | INS | $\leq$650°C | $\leq$650°C |
| $\delta(N–C\equiv N)_{cyanamide}$ | 700 | FTIR | not detected | $\geq$650°C |
| $\nu_3(CO_3)$ | ca 1450 | FTIR | $\leq$300°C | $\leq$600°C |
| $\delta(CH_2)_{lipids/proteins}$ | 1400–1500 | FTIR, INS | $\leq$300°C | $\leq$450°C |
| $\delta(NH) + \nu(CN)_{protein}$/amide II | 1550 | FTIR, INS | 200°C | $\leq$400°C |
| $\delta(C=O)_{protein}$/amide I | 1650 | FTIR, INS | $\leq$400–450°C | $\leq$650°C |
| $\nu(\equiv N)_{cyanamide}$ | 2009 | FTIR | not detected | $\geq$650°C |
| $\nu(CH_2)_{sp3}$ | 2980 | INS | 200–300°C | 200–450°C |
| $\nu(CH_2)_{sp2}$ | 3090–3100 | INS | 300–450°C | 450–650°C |
| $\nu(NH)_{protein}$/amide A | ca 3280 | FTIR, INS | $\leq$300°C | $\leq$550°C |
| $\nu(NH)/\nu(OH)_{HAp-disorg}$ | ca 3360 | FTIR | $\geq$300°C | $\geq$550°C |
| $\nu(OH)_{HAp}$ | 3570 | FTIR, INS | $\geq$300°C | $\geq$300°C |

conditions as compared to anaerobic settings. While protein is still detected at 550–600°C for the latter, it is very much reduced by 450 to 500°C in combusted bones. These plots clearly corroborate that the main heat-induced variations occur from 300 to 400°C for aerobically burned bones and from 450 to 500°C for anaerobically burned samples, in accordance with the corresponding vibrational profiles observed for both the organic and inorganic bone constituents.

## 4. Conclusion

Diagenetic alterations in bone can be accurately detected by vibrational spectroscopy. This provides reliable clues for precise characterization of the conditions that have shaped the samples (e.g. temperature, chemical composition of the surroundings, moisture, presence or lack of oxygen). Complementary FTIR and INS data were obtained for human bones heated between 200 and 650°C under different conditions regarding oxygen availability—aerobic and anaerobic (in a closed environment). Overall, for the samples burned in oxidizing settings the main spectral differences were detected between 300 and 400°C, while for those heated anaerobically the major changes took place from 450 to 500°C in agreement with the delayed heat effect on bone under reducing conditions (previously seen for higher burning temperatures [24,28]). The organic constituents of bone—lipids and proteins (mainly collagen)—were found to disappear at lower temperatures for aerobic heating conditions (at *ca* 300°C), while they lingered within the bone matrix to higher temperatures in anaerobic environments (*ca* 450–550°C). This is as previously reported for bovine [25] bones and in accordance with the fact that oxidation reactions also contribute to the degradation of lipids and polypeptides. Moreover, unsaturated carbon species (e.g. from lipidic constituents) were detected for the bones subject to these low and medium temperatures (less than or equal to 650°C), mainly for anaerobic conditions. Regarding the carbonates, they were also found to remain within the bone lattice up to higher temperatures for anaerobically burned samples—up to 450–550°C, when compared with 200–300°C for combusted bones. Additionally, cyanamide formation, previously only evidenced in bones burned anaerobically above 700°C [28], was found to take place by 650°C in reducing settings, but not at lower temperatures.

Coupling our new results with those for intense heating conditions (700–1000°C) [24,28], an interesting behaviour was evidenced regarding the temperature dependence of hydroxyapatite's OH librational mode for the wide 200–1000°C temperature range: in aerobic media the profile from 200 to 600°C was found to be inverse to that above 650°C, while in anaerobic settings a continuous increase in transition energy was observed for the whole temperature interval (200 to 1000°C).

The information obtained on the heat-induced variations in bone's organic and inorganic matrices for temperatures below 650°C, under both oxidizing and reducing conditions, complements former studies performed by the authors for higher burning temperatures. The complete data thus gathered, spanning from 200 to 1000°C (as well as for unburned samples), contribute to a better understanding of heat-induced diagenesis in bone. The spectral biomarkers already identified are able to be used in real scenarios, both forensic and archaeological. These allow the determination of the maximum heating temperatures of burned human skeletal remains (a persistent challenge up to this date). They also enable one to recreate the bone heating conditions—namely whether the specimens have been subject to cooking or to intense heating processes such as cremation or instrument manufacturing. They may also assist in the interpretation of bone mineralization processes that occur during fossilization, which may help to provide reliable information on the biology of extinct vertebrates [21].

Ethics. The human bones used in this work are part of a collection of human skeletons hosted at the Laboratory of Forensic Anthropology of the University of Coimbra. Research on this collection was granted to the authors by the Ethics Committee of the Faculty of Medicine of the University of Coimbra (reference number: CE_026.2016).

Data accessibility. The dataset (infrared and INS spectra) supporting this article is available from the Science and Technology Facilities data repository (eData) at: http://dx.doi.org/10.5286/edata/749.

Authors' contributions. M.P.M.M. conception of the study, data analysis and manuscript writing; L.A.E.B.C. conception of the study, sample preparation, FTIR experimental measurements and data analysis; D.G. sample preparation; E.C. provision of bone samples; S.F.P. INS experimental measurements, data analysis and manuscript writing. All authors have read and agreed to the published version of the manuscript.

Competing interests. The authors have no competing interests to declare.

Funding. The authors acknowledge financial support from the Portuguese Foundation for Science and Technology (UIDB/00070/2020). The STFC Rutherford Appleton Laboratory is thanked for access to neutron beam facilities (TOSCA/RB2010012 and MAPS/RB2010018, DOIs 10.5286/ISIS.E.RB2010012 and 10.5286/ISIS.E.RB2010018).

Acknowledgements. António Manuel de Carvalho Ferreira (Department of Chemistry, University of Coimbra) is acknowledged for planning and building the home-made chamber used for anaerobic burning of the bone samples under sealed conditions.

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
