## [Peer Review File · Royal Society Open Science]

Review History

RSOS-210774.R0 (Original submission)

Review form: Reviewer 1

Is the manuscript scientifically sound in its present form?

Yes

Are the interpretations and conclusions justified by the results?

Yes

Is the language acceptable?

Yes

Do you have any ethical concerns with this paper?

No

Have you any concerns about statistical analyses in this paper?

No

Recommendation?

Accept with minor revision (please list in comments)

Comments to the Author(s)

The manuscript reports the study about the effect of moderate heating on human bones components. The present manuscript explores the relatively low temperature range between 200 to 650 degree that never was studied in previous publications. The described results will be used as reference for future applications in forensic science, anthropology and cultural heritage.

Review form: Reviewer 2

Is the manuscript scientifically sound in its present form?

Yes

Are the interpretations and conclusions justified by the results?

Yes

Is the language acceptable?

Yes

Do you have any ethical concerns with this paper?

No

Have you any concerns about statistical analyses in this paper?

No

Recommendation?

Accept as is

Comments to the Author(s)

This is a well written, technically correct paper. The references are appropriate, the sample size and data collection are reasonable. The experimental parts were well and strictly analyzed, and the conclusion is sound and will be profitable for the readers. This is a useful addition to the archaeological/forensic osteology literature. I recommend that paper be accepted as is.

Decision letter (RSOS-210774.R0)

Dear Professor Batista de Carvalho,

Title: The Impact of Moderate Heating on Human Bones: An Infrared and Neutron Spectroscopy Study

Manuscript ID: RSOS-210774

Thank you for your submission to Royal Society Open Science. Your manuscript was sent for review and we have now received the report(s).

Having carefully evaluated your manuscript and the reviewer comments, it is a pleasure to accept your manuscript in its current form for publication in Royal Society Open Science. The chemistry content of Royal Society Open Science is published in collaboration with the Royal Society of Chemistry.

You can expect to receive a proof of your article in the near future. Please contact the editorial office (openscience@royalsociety.org) and the production office (openscience@royalsociety.org) to let us know if you are likely to be away from e-mail contact -- if you are going to be away, please nominate a co-author (if available) to manage the proofing process, and ensure they are copied into your email to the journal.

Thank you for your fine contribution. On behalf of the Editors of Royal Society Open Science and the Royal Society of Chemistry, I look forward to your continued contributions to the journal.

Yours sincerely,
Dr Ellis Wilde
Publishing Editor, Journals

On behalf of the Subject Editor Professor Anthony Stace and the Associate Editor Dr Nadia Martinez Villegas.

RSC Associate Editor
Comments to the Author:

The results presented in this paper are new data (in the 200-600 oC temperature range) and increase understanding on human bones that have been burned in a wider temperature range. Very interesting and well done piece of work.

RSC Subject Editor
Comments to the Author:
(There are no comments.)

Reviewer(s)' Comments to Author:

Reviewer: 1

Comments to the Author(s)

The manuscript reports the study about the effect of moderate heating on human bones components. The present manuscript explores the relatively low temperature range between 200 to 650 degree that never was studied in previous publications. The described results will be used as reference for future applications in forensic science, anthropology and cultural heritage.

Reviewer: 2

Comments to the Author(s)

This is a well written, technically correct paper. The references are appropriate, the sample size and data collection are reasonable. The experimental parts were well and strictly analyzed, and the conclusion is sound and will be profitable for the readers. This is a useful addition to the archaeologic/forensic osteology literature. I recommend that paper be accepted as is.